# Influence of Parenteral Nutrition Delivery Techniques on Growth and Neurodevelopment of Very Low Birth Weight Newborns: A Randomized Trial

**DOI:** 10.3390/medicina55040082

**Published:** 2019-03-28

**Authors:** Ilona Aldakauskienė, Rasa Tamelienė, Vitalija Marmienė, Inesa Rimdeikienė, Kastytis Šmigelskas, Rimantas Kėvalas

**Affiliations:** 1Department of Neonatology, Medical Academy, Lithuanian University of Health Sciences, LT-50009 Kaunas, Lithuania; ilona.aldakauskiene@lsmuni.lt (I.A.); rasa.tameliene@kaunoklinikos.lt (R.T.); 2Hospital of Lithuanian University of Health Sciences, LT-50009 Kaunas, Lithuania; mar.vitalija@gmail.com (V.M.); inesa.rimdeikiene@kaunoklinikos.lt (I.R.); 3Department of Rehabilitation, Medical Academy, Lithuanian University of Health Sciences, LT-50009 Kaunas, Lithuania; 4Department of Health Psychology, Medical Academy, Lithuanian University of Health Sciences, LT-47181 Kaunas, Lithuania; kastytis.smigelskas@lsmuni.lt; 5Health Research Institute, Medical Academy, Lithuanian University of Health Sciences, LT-47181 Kaunas, Lithuania; 6Department of Pediatrics, Medical Academy, Lithuanian University of Health Sciences, LT-50009 Kaunas, Lithuania

**Keywords:** preterm newborn, parenteral nutrition, peripherally inserted central venous catheter, peripheral venous catheter

## Abstract

*Background and Objectives:* In very low birth weight (VLBW) newborns, parenteral nutrition (PN) is delivered via a peripheral venous catheter (PVC), a central venous catheter (CVC), or a peripherally inserted central venous catheter (PICC). Up to 45% of PICCs are accompanied by complications, the most common being sepsis. A PVC is an unstable PN delivery technique requiring frequent change. The growth and neurodevelopment of VLBW newborns may be disturbed because of catheters used for early PN delivery and complications thereof. The aim of the conducted study was to evaluate the effect of two PN delivery techniques (PICC and PVC) on anthropometric parameters and neurodevelopment of VLBW newborns. *Materials and Methods:* A prospective randomized clinical trial was conducted in VLBW (≥750–<1500 g) newborns that met the inclusion criteria and were randomized into two groups: PICC and PVC. We assessed short-term outcomes (i.e., anthropometric parameters from birth until corrected age (CA) 36 weeks) and long-term outcomes (i.e., anthropometric parameters from CA 3 months to 12 months as well as neurodevelopment at CA 12 months according to the Bayley II scale). *Results:* In total, 108 newborns (57 in the PICC group and 51 in the PVC group) were randomized. Short-term outcomes were assessed in 47 and 38 subjects, and long-term outcomes and neurodevelopment were assessed in 38 and 33 subjects of PICC and PVC groups, respectively. There were no differences observed in anthropometric parameters between the subjects of the two groups in the short- and long-term. Mental development index (MDI) < 85 was observed in 26.3% and 21.2% (*p* = 0.781), and psychomotor development index (PDI) < 85 was observed in 39.5% and 54.5% (*p* = 0.239) of PICC and PVC subjects, respectively. *Conclusions:* In the short- and long-term, no differences were observed in the anthropometric parameters of newborns in both groups. At CA 12 months, there was no difference in neurodevelopment in both groups.

## 1. Introduction

Significant delay in weight gain is observed at an early postnatal stage among very low birth weight (VLBW) newborns who survive [1]. This is caused by morbidities (respiratory distress syndrome, patent ductus arteriosus, bronchopulmonary dysplasia, late-onset neonatal sepsis, necrotic enterocolitis, intraventricular hemorrhage grade III-IV) [2], as well as the calorie and protein deficit during the first weeks. Early parenteral nutrition (PN) reduces this delay in weight gain [3]. In newborns, PN may be delivered via a peripheral venous catheter (PVC) or a central venous catheter (CVC). Among CVCs, peripherally inserted central venous catheters (PICCs) are most common at a neonatal age. PN recommendations and protocols indicate central and peripheral venous catheters as possible techniques of catheter insertion, prioritizing central venous catheters [4,5,6,7,8]. The techniques have both similarities and differences. Both PICCs and PVCs are brought in via a peripheral vein; however, the position of the end of the catheter differs, as does the duration of use and complications thereof. The average duration of use of a PICC is between 7 and 14 days [9,10]; meanwhile, the average use of a PVC is 3 days [11]. Uninterrupted PN using a PICC or a PVC requires different numbers of painful catheter insertion procedures. Frequently changing catheters is likely to have an effect on the amount of nutrients taken in and weight gain in newborns [12]. The pain experienced by pre-term newborns increases the risk of developmental disorders at a later stage [13,14]. Catheters used for PN may cause numerous complications and influence long-term outcomes in newborns. According to the literature, from 2.9% [15] to 45.9% [16] of all used PICCs are changed or removed following a complication. Central-line-associated blood stream infection (CLABSI) is the most common complication of PICCs (up to 19%) [17]. It is claimed that the use of a PICC may result in embolization, cardiac tamponade, or pleural effusion. Although rare, these complications are potentially life-threatening [15,18]. According to the data of various studies, 41–50% of PVCs are removed or changed as a result of complications, the most common being local infiltration, extravasation, or occlusion [11,19]. Some studies report the rate of PVC-associated sepsis to be 0.1% [19]. Based on the findings of other studies, this technique yields a similar risk of sepsis as in the use of PICCs [20,21]. In addition to PVC, both PN and PICC are considered to be risk factors for sepsis [20,22]. Sepsis is associated with a greater risk of a delay in weight gain, neurocognitive disorders, and death in VLBW newborns [23,24,25,26].

A number of Cochrane meta-analyses (2004, 2007, 2015) were conducted, which focused on the differences of PICCs and PVCs used for PN [27,28,29]. One of the objectives of the meta-analyses was to determine the effect of a PN delivery technique on the growth and development of newborns. The studies generalize the rate of complications in the groups of subjects with different catheters as well as the number of catheters used for PN and other parameters. None of the studies have provided data about the growth and psychomotor neurodevelopment of newborns [29].

Modern neonatology aims to use the least-invasive treatment techniques in order to ensure the needs of VLBW newborns are met, and to achieve the best outcomes of their growth and neurodevelopment. We conducted a randomized trial to test the hypothesis that early PN using a PVC will ensure similar nutrition, growth, and development of newborns as PN using a PICC.

The aim of the study was to assess the effect of two PN delivery techniques (PICC and PVC) on anthropometric parameters and psychomotor neurodevelopment of VLBW newborns.

## 2. Materials and Methods

### 2.1. Study Population

The prospective randomized clinical trial was conducted at the Hospital of the Lithuanian University of Health Sciences Kauno klinikos. The protocol was approved and permission No. BE-2-3 was granted by Kaunas Regional Biomedical Research Ethics Committee. The trial was conducted at the Department of Neonatology of the Hospital of the Lithuanian University of Health Sciences Kauno klinikos between 1 January 2014 and 31 March 2017. The subjects were premature neonates who were born at the Hospital of the Lithuanian University of Health Sciences Kauno klinikos and were treated post-birth in the Neonatal Intensive Care Unit.

The following inclusion criteria had to be met:Birth weight ≥750–<1500 g, anticipated PN duration no less than 5 days and written consent of both parents to participate in the trial.

The exclusion criteria were as follows:Newborns with chromosomal diseases, multiple dysplasia, coagulation disorders, and skin anomalies preventing catheterization; newborns in demand of central vein catheterization observed post-birth due to administered inotropic medicines; and newborns whose parents refused to participate in the trial.

During the trial period, 165 newborns (birth weight ≥750–<1500 g) born at the Hospital of the Lithuanian University of Health Sciences Kauno klinikos were hospitalized at the Neonatal Intensive Care Unit. Of them, 108 (65.5%) met the eligibility criteria (Figure 1).

The main reason for exclusion was parents’ refusal (17.8%). During the trial, some subjects were excluded because of exclusion criteria that occurred in the course of the trial. The most common cause of exclusion was PN shorter than 5 days (65.2%). Other reasons were less common. The analyzed groups were not equal in terms of the number of the subjects due to blind randomization and a different exclusion rate.

During the trial, five subjects (5.9%) were transferred from the group assigned by blind randomization to another one due to clinical parameters. Three subjects (7.9%) were re-assigned from the PVC group to the PICC group. In all the cases, the reason for change of the group was a severe condition in a newborn requiring central venous catheterization (multiple PVC catheterizations because of fragile blood vessels). Two subjects (4.3%) were re-assigned from the PICC group to the PVC group. In both cases, the reason for the change of group was PICC-related complications, and based on the clinical condition it was decided not to insert a new central venous catheter. The data analysis of the subjects who changed from the initially assigned group were assessed until the day of group change. Later, the existing data were not included in the general analysis (i.e., were censored). The decision was made based on the fact that both interventions may have influenced the outcomes, and it was impossible to determine which influence was greater.

### 2.2. Intervention

When a neonate complying with the inclusion criteria was born, blind randomization along with prepared numbered questionnaires and catheterization were conducted during the first 24 h. Until that, regular treatment was applied to all neonates. After blind randomization, catheterization was performed, and any unsuitable catheters were removed. Catheters were inserted following the procedural protocols of the hospital. In the study, 24G cannulas (B. Braun Melsungen AG, Germany) and 28G (1FR) PICCs (Premicath Vygon) were used. The position of a PICC was confirmed by X-ray and, if needed, may have been corrected. PN was started on the first day of the neonate’s life, immediately after insertion of a vein catheter. Both subject groups were administered the same individual PN, following the protocol of PN and infusion therapy approved at the hospital. The components of the PN solution were as follows: dextrose, amino acids (Vaminolact), electrolytes, trace elements (Peditrace), a phosphorus preparation (Glycophos), lipid emulsions (Smoflipid), and vitamins (water-soluble Soluvit and fat-soluble Vitalipid). On the first days, the amount of nutrients administered along with PN was increased; later, it was reduced in proportion to increasing amounts obtained by enteral nutrition (EN). In both subject groups, EN was started as early as possible and was increased depending on the clinical condition of neonates. PN was finished and the chosen catheter was removed upon full EN achievement. Catheters were also removed or changed in the case of complications.

### 2.3. Data Collection

The weight of newborns was measured daily using digital scales; the length and the head circumference were measured using a flexible strip once a week. The weight was recorded daily until PN was administered and weekly when PN was completed. After discharge, the anthropometric data of the subjects were assessed at corrected age (CA) 3, 6, and 12 months. Additionally, at CA 12 months, assessment of mental and motor neurodevelopment of the subjects was conducted according to the Bayley II scale by a trained specialist.

### 2.4. Outcome Measures

Short-term anthropometric outcomes using the Fenton dataset [30];Long-term anthropometric outcomes using WHO standards [31];Neurodevelopment outcomes using the Bayley II scale.

### 2.5. Statistical Methods

The data were processed using MS Excel 2010 and IBM SPSS Statistics 20 software (IBM Corp, Armonk, NY, USA). The univariate analyses included calculation of means (±SD), medians (with interquartile ranges), and percentage distributions. A comparison of dichotomous indicators between the groups was performed using the chi-squared test or the Fisher’s exact test if assumptions for the former were not met. The analysis of anthropometric indicators was based on z-scores—standardized measures for the development of newborns. A comparison of z-scores between the study groups was performed using the Student’s *t*-test with respect to the Levene’s test for equality of variances. The variables with non-Gaussian distributions were compared using Mann–Whitney test. The statistical significance level was set at *p* < 0.05.

## 3. Results

### 3.1. Baseline Characteristics of Subjects

The mean gestational age, anthropometric data (weight, length, and head circumference), and the Apgar scores in both subject groups did not differ (Table 1).

The mean PN duration as well as the administered nutrients and energy values were not statistically significantly different between the groups. There were also no statistically significant differences in morbidity by neonatal diseases and mortality following birth until discharge between the groups (Table 2).

### 3.2. Short-Term Anthropometric Outcomes

The comparison of z-scores of the weight, length, and head circumference of the subjects at CA 7, 14, 21, and 28 days as well as 36 weeks did not show any statistically significant differences between the groups (Figure 2).

### 3.3. Long-Term Anthropometric Outcomes

The analysis of the dynamics of anthropometric parameters in the long-term showed no statistically significant differences in the z-scores of the weight, length, or head circumference at CA 3, 6, and 12 months. From CA 3 to 12 months, the z-scores of the weight, length, and head circumference increased in both studied groups (Figure 3).

### 3.4. Neurodevelopment Outcomes

The analysis of mental development outcomes demonstrated that the majority of the subjects in both groups were at normal and very good mental development: 73.7% in the PICC group and 78.8% in the PVC group, respectively (*p* = 0.781). The rate of a significant delay in mental development in the PICC and PVC groups was 10.5% and 0%, respectively (*p* = 0.118). The total delay in mental development was 26.3% vs. 21.2% (*p* = 0.781) in the PICC and PVC groups, respectively. There was no statistically significant difference in mental development between the groups (Table 3).

The analysis of motor development outcomes showed normal and very good motor development in 60.5% and 45.5% (*p* = 0.239) of the PICC and PVC subjects. The rate of a significant delay in motor development in the PICC and the PVC groups was 13.2% and 0%, respectively (*p* = 0.057). A total delay in motor development was observed in 39.5% and 54.5% (*p* = 0.239) of the subjects in the PICC and the PVC groups, respectively. There was no statistically significant difference in motor development between the groups (Table 4).

## 4. Discussion

The growth and psychomotor neurodevelopment of preterm newborns depend on a number of factors. The duration of PN, the amounts of nutrients taken in during PN, the catheters used for PN and the complications thereof, as well as painful catheter insertion procedures experienced by newborns also affect the growth and neurodevelopment of preterm newborns at an early and later phase. Studies that have focused on the comparison of PICCs and PVCs assessed the number of used catheters, the rate of complications, and other parameters [21,29,32,33]. We did not find any studies that analyzed the effect of PN delivery technique on the growth and development of preterm newborns. One study has determined a greater difference between planned and taken-in nutrients when PVCs were used for PN in comparison with PICCs; however, the effect of such a difference on the growth of newborns was not assessed [12]. It may be claimed that our research is one of the first to assess the effect of PN delivery technique on the growth and neurodevelopment of VLBW newborns.

Literature sources claim that the growth of VLBW newborns at an early age is insufficient. The z-scores of the weight, length, and head circumference gradually decrease from birth until discharge. Recent data show that the z-score of weight in VLBW newborns upon discharge was -1.54 ± 0.75 [34], −1.46 [35], and that of newborns at a gestational age of <34 weeks was −0.97 ± 0.7 [36]. In our study, we determined that both subject groups were equal according to the baseline characteristics and further data. We analyzed the weight, length, and head circumference from birth until CA 36 weeks (CA 34, 35 weeks if discharged earlier) and determined that the z-scores of all anthropometric data from birth until CA 36 weeks decreased in both the PVC and the PICC groups and were not significantly different between the groups. The z-scores of the weight of newborns at CA 36 weeks were −1.44 and −1.12 (*p* = 0.968) in the PVC and the PICC subjects, respectively. The z-scores of the length and head circumference were similar and were not significantly different between the groups. Our results do not contradict the data reported in literature. Our results demonstrate that the growth of VLBW newborns at an early age is insufficient regardless of the PN delivery technique.

The long-term growth data of VLBW newborns as reported in the literature range between significantly to minimally delayed performance. Tendencies show the maximum decrease in z-scores of anthropometric parameters at an early age, and they are greater in the long-term, which shows a decrease in delayed performance or disappearance overall. It has been reported that newborns of gestational age of <33 weeks had the z-score of −0.8 and −0.2 for weight at CA 4 months and 12 months, respectively [37]; meanwhile the z-score of VLBW newborns at CA 3 months and 12 months was −2.4 ± 1.3 and −1.2 ± 1.3, respectively [38]. During our study, we determined that the z-score dynamics from CA 3 months to 12 months in the PVC and the PICC subjects were from −0.05 to 0.6 and −0.89 to −0.17 (*p* = 0.1 and *p* = 0.3), respectively. There was a similar change in the z-scores of the length and the head circumference, and no difference between the groups was observed. The comparison of our data with those reported in the literature demonstrate that our data on the long-term development of newborns until CA 12 months do not contradict the results of other studies. The growth observed in both groups of newborns was not dependent on the PN delivery technique.

Few methodologies of neurodevelopment assessment are used worldwide and as the age of newborns differs upon assessment, comparison of data is complicated. When the Bayley II scale has been employed, significantly delayed mental development at CA 12 months has been observed in 12% and 12.89% of newborns, and significantly delayed motor development has been observed in 17.8% of VLBW newborns [39,40]. Fairly delayed mental development has been observed in 29.3% and 25.1% of newborns, and fairly delayed motor development has been observed in 26.8% of VLBW newborns [39,40]. When the Bayley III scale has been used to measure performance, significantly delayed cognitive development has been observed in 8.5% and significantly delayed motor development in 7.6% of subjects [41]. With the HMURGC (Havana’s Dr. Ramón González Coro University Maternity Hospital) protocol, fairly delayed psychomotor development has been determined in 25.9% and significant delay in 5.2% of subjects [42]. In our study, we assessed neurodevelopment according to the Bayley II scale. At CA 12 months, the majority of the subjects in the PVC and the PICC groups demonstrated normal or very good mental and motor development, and there was no difference between the groups (*p* = 0.781, *p* = 0.239). Delayed mental development was determined in 21.2% and 26.3% (*p* = 0.781) of the PVC and PICC subjects, respectively. Delayed motor development was determined in 54.5% and 39.5% (*p* = 0.239) of the subjects in the PVC and PICC groups, respectively. Although significantly delayed mental and motor development was more common in the PICC group in comparison with the PVC group, there was no statistically significant difference between the groups. Our findings on psychomotor neurodevelopment parameters do not contradict those reported in the literature. At CA 12 months, psychomotor neurodevelopment of VLBW newborns was not dependent on the PN delivery technique.

The findings that we obtained on neurodevelopment parameters do not contradict the data reported in the literature. At 12 months, we compared the influence of two PN delivery techniques on the anthropometric parameters and psychomotor neurodevelopment of VLBW newborns, and did not determine any difference between the groups. Neither of the methods showed an advantage. As PVCs are less invasive, cause no life-threatening complications, are cheap, and require no specific technical preparation, this technique should be the first choice in early PN delivery.

### Limitations

One limitation of our study may have been a relatively small sample size. Our sample size was limited due to natural restrictions in absolute numbers of the target group (VLBW newborns) and, additionally, the eligibility criteria. Nonetheless, we made post-hoc power calculations for the most clinically significant outcomes—mental and psychomotor neurodevelopment. We found that the power was above 95% for the significantly delayed performance, while for the delayed performance it was below 50%. The latter power can be regarded as low; however, it should be noted that the differences between treatment arms were ambiguous (psychomotor performance was better in the PICC group while mental performance was better in the PVC group), indirectly suggesting a certain level of treatment equivalence. Another limitation in our study was that we did not use an intention-to-treat approach. The reason for this was that five children dropped out from the study due to switching the treatment group within the first days of treatment. The cause of change was the indication to withdraw the intervention based on objective criteria. We decided to withdraw these cases from the calculations, because we assumed that several days in the beginning of the intervention were not likely to affect the outcomes several months or one year later.

## 5. Conclusions

No differences in the growth of newborns until CA 36 weeks were determined between the groups. At an early stage, age-dependent progressing growth delay was observed in both subject groups. There were also no observed differences in the growth of newborns in the long term. Both groups demonstrated a gradually increasing growth of newborns upon discharge. The rate of mental and motor development did not differ. Although a greater rate of significantly delayed mental and motor development was observed in the PICC group, no statistically significant difference was determined between the groups.

## Figures and Tables

**Figure 1 medicina-55-00082-f001:**
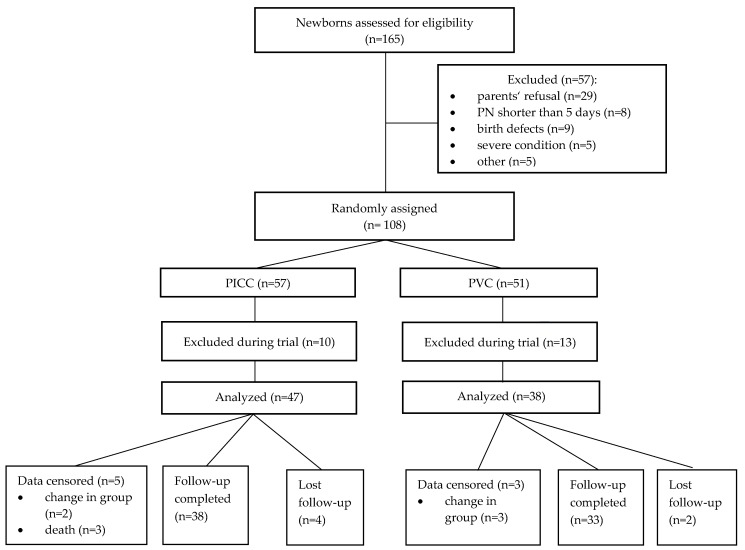
Study design showing the flow of participants throughout the trial. PN: parenteral nutrition; PICC: peripherally inserted central venous catheters; PVC: peripheral venous catheter.

**Figure 2 medicina-55-00082-f002:**
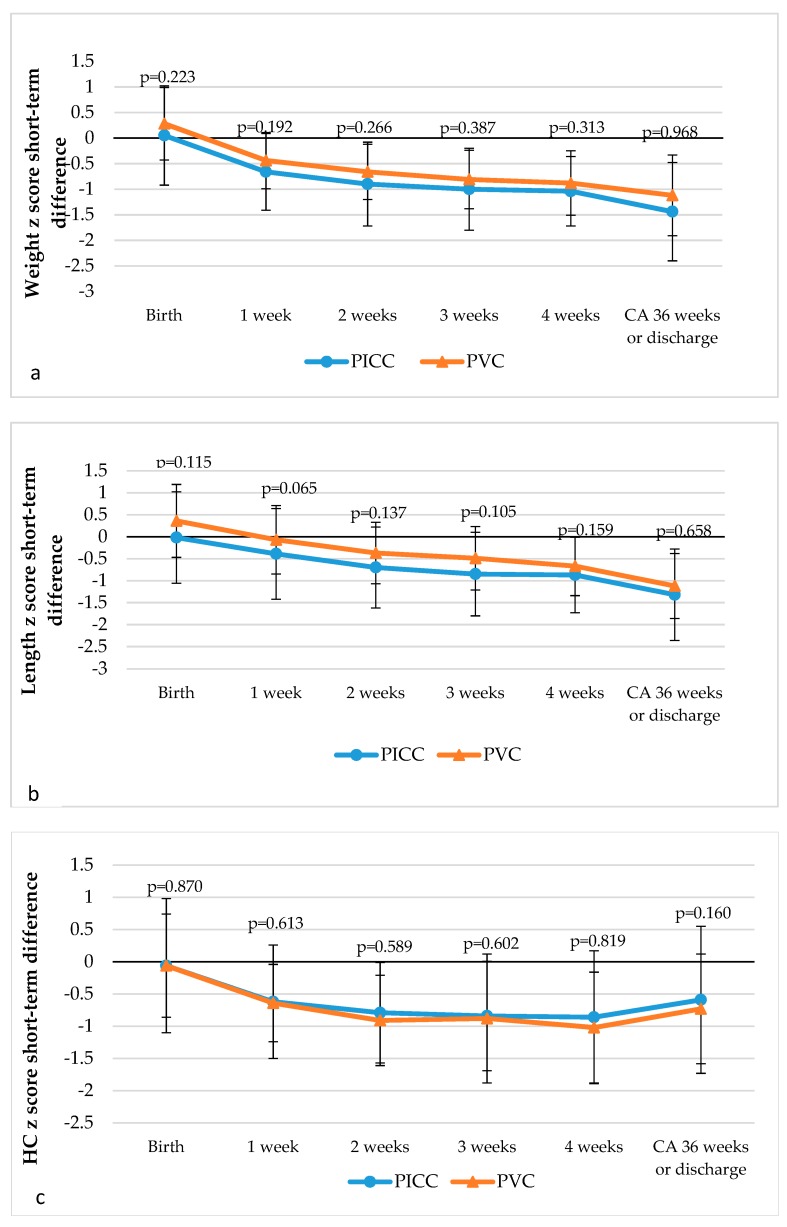
Dynamics of weight (**a**), length (**b**), and head circumference (**c**) z-scores from birth until corrected age (CA) 36 weeks (34 or 35 weeks, in case of earlier discharge). HC: head circumference.

**Figure 3 medicina-55-00082-f003:**
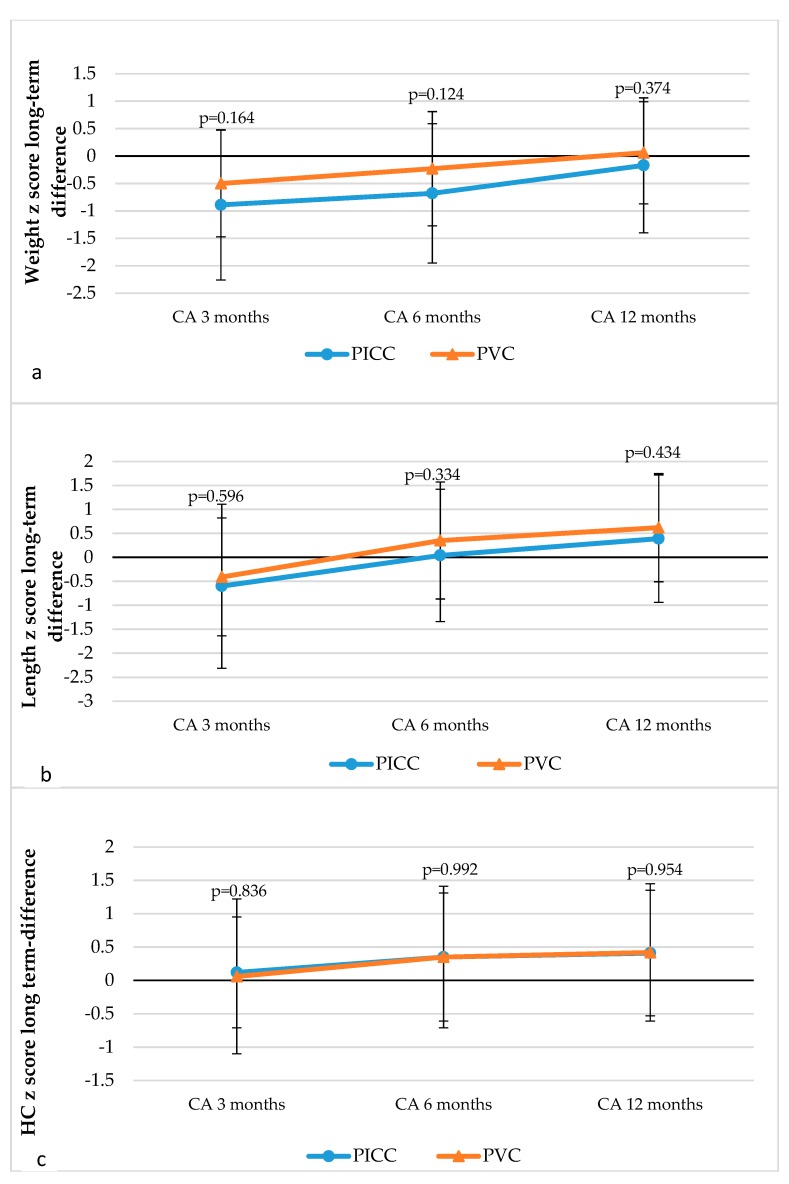
Dynamics of weight (**a**), length (**b**), and head circumference (**c**) z-scores from CA 3 months to 6 months.

**Table 1 medicina-55-00082-t001:** The main characteristics of subjects at the beginning of the trial.

Characteristics	PICC Group (*n* = 47)	PVC Group (*n* = 38)	*p*
Gestation, median (IQR), weeks	28.0 (27; 30)	28.5 (27; 29)	0.936
Birth weight, median (IQR), g	1086 (900; 1270)	1196 (987.75; 1280.5)	0.265
Sex, *n* (%)			0.355
Male	27 (57.4)	18 (47.4)	
Female	20 (42.6)	20 (52.6)	
SGA, *n* (%)	5 (10.6)	1 (2.6)	0.218
Length, median (IQR), cm	37 (35; 38)	37 (36; 39)	0.183
Head circumference, median (IQR), cm	26 (24; 27)	26 (24.75; 27)	0.875
Apgar 1 min, median (IQR)	7 (5; 8)	7 (5; 7)	0.356
Apgar 5 min, median (IQR)	8 (7; 8)	8 (7; 8)	0.507

SGA: small for gestational age.

**Table 2 medicina-55-00082-t002:** PN characteristics, morbidity, and mortality from birth before discharge of subjects (data censored).

Characteristics	PICC Group (*n* = 45)	PVC Group (*n* = 38)	*p*
PN duration, median (IQR), days	8 (6; 9)	7(6; 9)	0.331
Nutrients			
Amino acids, median (IQR), g/kg/d	2.51 (2.24; 2.76)	2.45 (2.20; 2.81)	0.593
Carbohydrates, median (IQR), g/kg/d	7.29 (6.46; 8.20)	7.25 (5.88; 8.41)	0.309
Lipids, median (IQR), g/kg/d	1.93 (1.63; 2.15)	1.93 (1.60; 2.04)	0.919
Energy, median (IQR), kcal/kg/d	53.1 (47.3; 58.4)	51.2 (44.9; 57.8)	0.536
Morbidity and mortality, *n* (%)			
RDS	41 (91.1)	33 (94.3)	0.691
BPD	6 (13.3)	8 (22.9)	0.266
PDA	22 (48.9)	18 (51.4)	0.822
Early onset sepsis	9 (20.0)	11 (31.4)	0.242
Late onset sepsis	8 (17.8)	7 (20.0)	0.801
IVH III	1 (2.2)	1 (2.9)	1.000
IVH IV	4 (8.9)	0 (0)	0.127
PVL	14 (30.4)	10 (28.6)	1.00
Mortality before discharge	3 (6.4)	0 (0)	0.25

RDS: respiratory distress syndrome; BPD: bronchopulmonary dysplasia; PDA: patent ductus arteriosus; IVH: intraventricular hemorrhage; PVL: periventricular leukomalacia.

**Table 3 medicina-55-00082-t003:** Mental development according to the Bayley II scale at CA 12 months.

MDI	PICC, *n* (%)	PVC, *n* (%)	Fisher *p*	χ^2^
<70 (significant delay)	4 (10.5)	0 (0.0)	0.118	3.68
<85 (total delay)	10 (26.3)	7 (21.2)	0.781	0.25

MDI: mental development index.

**Table 4 medicina-55-00082-t004:** Motor development according to the Bayley II scale at CA 12 months.

PDI	PICC, *n* (%)	PVC, *n* (%)	Fisher *p*	χ^2^
<70 (significant delay)	5 (13.2)	0 (0.0)	0.057	4.67
<85 (delay)	15 (39.5)	18 (54.5)	0.239	1.61

PDI—psychomotor development index.

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
