# Peer review of "Influence of Parenteral Nutrition Delivery Techniques on Growth and Neurodevelopment of Very Low Birth Weight Newborns: A Randomized Trial"

_medicina, 2019, doi:10.3390/medicina55040082_

Reviewer 1 Report

Thank you for revising the manuscript. I agree with the changes and recommend it for publication.

Reviewer 2 Report

Authors completed changes proposed by revision and the work has been improved in my opinion. Perhaps it would need minor grammar checking.

My recommendation is to accept the work.

Thank you to authors for fast and accurate work.

This manuscript is a resubmission of an earlier submission. The following is a list of the peer review reports and author responses from that submission.

Round  1

Reviewer 1 Report

The authors have attempted to look at potential differences in short- and long-term growth outcomes and neurodevelopment between two methods of delivery of parenteral nutrition. I am not sure that the hypothesis is fully justified, although a few weak arguments are made.

There are several major comments to make:

1) No a-priori sample size calculation was attempted. Even if there are no previous studies to use for estimation, a sample size calculation should be attempted. If this is not possible, then this study should be presented as a pilot study.

2) There are several sources of potential bias, some more serious than others. There is no mention of allocation concealment, details of performance, and a serious risk of attrition bias. An intention-to treat analysis should strongly be considered between the groups, and protocol violations should not be discarded. In this case, it has unbalanced the groups.

3) There are several potential confounders which need to be discussed: actual duration of PN exposure, quality of PN solution and its composition, other sources of nutrition being used (feeds) and their type, any other co-existing morbidity, etc. If these are not accounted for or presented, it becomes impossible to judge the results.

Minor comments

1) Why were Fenton's charts used for the short-term outcomes? Are these reflective of babies in Lithuania?

2) Some results have been mentioned in the methods section; these should be separated.

Reviewer 2 Report

Ref. medicina-437944

Influence of parenteral nutrition delivery techniques on growth and neurodevelopment of very low birth weight newborns: a randomized trial

Review

The objective of the study was to assess the effect of two PN delivery techniques (PICC vs. PVC) on anthropometric parameters and psychomotor neurodevelopment of LBW newborns with a 12-month follow-up.

a)       Please update reference 7 to the last guidelines in Clin Nutr 2018, 37 (6), Part B: 2303-2430. An issue dedicated to pediatric nutrition guidelines. Specifically, i.e.: (Kolacek, Puntis, Hojsak, & nutrition, 2018)

b)      Apgar score should be provided

c)       Since the study compared two type of venous access, authors should provide information of:

a.       what type of devices they used (material, number of lumens, …)?

b.       what type of insertion was performed (ultrasound guided, radiologically guided…)?

c.       site of insertion

d.       This information is important since venous access techniques have some diversity amongst hospitals (Taylor, McDonald, & Tan, 2014). The study relevance will be different depending on those used.

d)      Table 1 has to be moved to Results

e)      Received both cohorts the same type of nutrients i.e.: fat emulsion, amino acid solution? Received the first patients the same type of nutrients than the most recent patients when the study recruitment lasted 3 years?

f)        Did the patients received the same vitamins and trace elements as part of PN?

g)       Paragraph corresponding to lines 131 – 135 has to be moved to Results.

h)      Have the authors recorded the interruptions of PN delivery in both cohorts? Were they different?

i)        SD should be provided when appropriated, i. e. nutrients administered, graphic values,..

j)        Provide absolute values in addition to %, i.e.: % in paragraph beginning in line 177.

k)       Line 259: A limitation…

References

Kolacek, S., Puntis, J. W. L., Hojsak, I., & nutrition, E. E. E. C. w. g. o. p. p. (2018). ESPGHAN/ESPEN/ESPR/CSPEN guidelines on pediatric parenteral nutrition: Venous access. Clin Nutr, 37(6 Pt B), 2379-2391. doi:10.1016/j.clnu.2018.06.952

Taylor, J. E., McDonald, S. J., & Tan, K. (2014). A survey of central venous catheter practices in Australian and New Zealand tertiary neonatal units. Aust Crit Care, 27(1), 36-42. doi:10.1016/j.aucc.2013.11.002

Reviewer 3 Report

My overall concern about this paper is that it fails to address one of the primary benefits of using a PICC line - the ability to provide higher concentrations of sugar, protein, and minerals to provide better nutrition. Instead they gave essentially the same nutrition to both groups, and then were "surprised" that they grew the same and had the same neurodevelopmental outcomes. If they truly want to prove that kids getting parenteral nutrition through a PIV turn out the same, they need to compare proper nutrition for both groups.  In the US, our kids with a PICC get 4 g/kg of protein and 90+ kcal/kg; almost twice as much as in their study.